# Evaluate the Effectiveness of Outpatient Parenteral Antimicrobial Therapy (OPAT) Program in Saudi Arabia: A Retrospective Study

**DOI:** 10.3390/antibiotics11040441

**Published:** 2022-03-24

**Authors:** Haneen J. Al Shareef, Adnan Al Harbi, Yasser Alatawi, Ahmed Aljabri, Mohammed A. Al-Ghanmi, Mohammed S. Alzahrani, Majed Ahmed Algarni, Attiah Khobrani, Abdul Haseeb, Faisal AlSenani, Mahmoud E. Elrggal

**Affiliations:** 1Clinical Pharmacy, King Abdullah Medical City, Makkah 21955, Saudi Arabia; haneen.albarakati@gmail.com (H.J.A.S.); alganmim@kamc.med.sa (M.A.A.-G.); khobrani.a@kamc.med.sa (A.K.); 2College of Pharmacy, Umm Al Qura University, Makkah 21955, Saudi Arabia; assharbi@uqu.edu.sa (A.A.H.); amhaseeb@uqu.edu.sa (A.H.); fssenani@uqu.edu.sa (F.A.); 3Department of Pharmacy Practice, Faculty of Pharmacy, University of Tabuk, Tabuk 71491, Saudi Arabia; yasser@ut.edu.sa; 4Department of Pharmacy Practice, Faculty of Pharmacy, King Abdulaziz University, Jeddah 21589, Saudi Arabia; amaljabri@kau.edu.sa; 5Clinical Pharmacy Department, College of Pharmacy, Taif University, Taif 21944, Saudi Arabia; m.s.alzahrani@tu.edu.sa (M.S.A.); m.alqarni@tu.edu.sa (M.A.A.)

**Keywords:** antimicrobials, parenteral, outpatients, antimicrobial stewardship

## Abstract

(1) Background: Outpatient parenteral antibiotic therapy (OPAT) is a well-established and cost-effective measure that improves the efficient use of healthcare resources and increases bed availability. Only limited published data is available to illustrate OPAT implementation and outcomes in Saudi Arabia. The main objective of this study was to evaluate the effectiveness of OPAT in a tertiary center in Saudi Arabia. (2) Methods: In this retrospective study, clinical charts of enrolled patients were reviewed in a tertiary care center from the initial month of November 2017 to March 2020. All admitted patients with a central line and who enrolled in the OPAT of the hospital during this study period were included. The primary outcome was the 30-days readmission rate of OPAT patients. Secondary outcomes were factors associated with OPAT failure. Descriptive analysis of the data was used to express the results. (3) Results: We enrolled 90 patients; 54 (60%) were male; the mean age was 55.16 (±17.7) years old. The mean duration of the antimicrobial treatment was 21.9 (+24.6) days. All patients completed the intended course of therapy. Ertapenem was the most frequently used antimicrobial (43%), followed by vancomycin (11.2%). Urinary tract infections (UTIs) are some of the most common bacterial infections in 25 patients (26.9%), followed by osteomyelitis in 16 patients (17.2%). Extended-spectrum beta-lactamase *E.coli* was the highest common isolated microorganism (44.9%), followed by methicillin-resistant Staphylococcus aureus MRSA (16.9%). The readmission to the hospital during therapy was required for 12 patients (13.3%). Shifting from hospital care to OPAT care resulted in cost savings of 18 million SAR in the overall assessment period and avoided a total of 1984 patient days of hospitalization. (4) Conclusion: The findings have shown that OPAT therapy was effective with minimum hospital readmissions and therapy complications. OPAT programs can reduce healthcare costs and should be integrated into practice.

## 1. Introduction

In Saudi Arabia, there is emerging heed towards ambulatory and outpatient healthcare services for conditions traditionally treated in hospitals in recent years [1]. The primarily driven factors are health care costs, patient preference, and the increase of hospital-acquired infection [2]. One such service applicable for such a transformation from inpatient to outpatient is antimicrobial therapy for patients who require parenteral antimicrobial therapy but are stable and do not require continuous healthcare supervision [3]. Outpatient parenteral antibiotic therapy (OPAT), also known as outpatient intravenous antibiotic therapy (OPIVAT), can be used to treat non-life-threatening infections and was first explained in 1974 by Rucker and Harrison [4,5]. OPAT is a well-established and approved outpatient healthcare service. OPAT offers several advantages over the traditional management of infectious diseases for both patients and the hospital. The patient’s ability to return home accelerates physical and psychological recovery and improves patient satisfaction [6]. A recent study by Yadav et al. found a meaningful impact of the implementation of OPAT clinical on return to emergency departments with limited therapy failure [7]. In addition, patients avoid an unnecessary hospital stay, which might increase the risk of nosocomial infection [8]. From the hospital perspective, OPAT is cost-effective, improves the efficient use of health care resources, and increases bed occupancy [9,10]. The OPAT program can be provided through two models: an OPAT infusion clinic (IC-OPAT), in which patients receive antimicrobial therapy in the Day Care Unit and a OPAT Home infusion Services (H-OPAT) [11].

The OPAT program was established in November 2017 in a tertiary care center in Saudi Arabia, in which both models were adopted. The program objectives included: (1) to provide a standardized approach for the administration of parenteral antimicrobials in an outpatient setting for patients requiring prolonged antimicrobial therapy; (2) to ensure safe and effective parenteral antimicrobial treatments within a community setting; (3) to decrease microorganism resistance burden; (4) to assess the already discharged patients from other hospitals that solely requires parenteral antimicrobial therapy; (5) to reduce the length of hospitalization and cost of health care services, and (6) to improve bed turnover time with a bed management system [11].

The implementation of OPAT has been extensively studied. Several guidelines, such as patient recommendations, antibiotic selection, monitoring, and follow-up procedures, have been published for adult and pediatric populations [11,12]. Patient monitoring is crucial for the successful implementation of OPAT because the risk of treatment failure, complications, and adverse events have been reported [13]. These outcomes might lead to a readmission or an emergency visit during OPAT. Evidence has identified several factors that contribute to the increase in readmission rates. For instance, patient age, type of antimicrobial resistance, and less than optimal monitoring can lead to readmission and increase the risk of morbidity and mortality [14].

There are very few centers in Saudi Arabia that utilize the OPAT program. Moreover, there is limited published data about OPAT implementation and outcomes in the kingdom. Therefore, this study aimed to estimate the 30 days readmission rate of OPAT patients and identify factors associated with OPAT failure.

## 2. Results

This study includes 90 patients who were discharged from a tertiary care center from November 2017 to March 2020 to receive outpatient IV antimicrobial therapy. Most of the patients were male (60%), and their mean age was 55 ± 17.7 years old. The most common infections which compelled patients to take OPAT were urinary tract infection (26.9%), osteomyelitis (17.2%), bacteremia (16.1%), hepatic micro abscess (5.4%), and diabetic foot (2.2%) in overall percentage (Table 1).

The mean length of OPAT duration was 21.9 ± 24.6 days. The infectious disease diagnosis with long-term IV administration of antimicrobial agents were tuberculosis for 72.25 ± 70 days and osteomyelitis for 42 ± 29.2 days. The most common isolated pathogens were ESBL *E. coli* (44.9%) and *MRSA* (16.9%) (Table 2). The most frequently used antibiotics were ertapenem (43%), vancomycin (11.2%), and cloxacillin/flucloxacillin (9.7%) (Table 3). The line-related minor complications were noted in 2 of 90 (2.2%) episodes. Only a few hospital readmission complications with no undeviating relation to the OPAT program were reported in 10 of 90 (11.2%). There were no severe complications that required hospital readmission in this study period (Figure 1).

## 3. Discussion

OPAT is mainly about providing better patient-centered healthcare services nearer to home while preventing adverse events and other risks associated with hospitalization. The findings have shown that OPAT therapy was effective with minimum hospital readmissions and therapy complications. The OPAT program can reduce healthcare costs and should be integrated into practice. While OPAT has been expanding in Saudi Arabia during the last few years because of its clinical benefits of reducing the length of hospital stay, there is a lot of variation in the availability of OPAT. In the healthcare system, OPAT is recognized as an extra cost, one of the main barriers to implementing this program. Therefore, this study was conducted to evaluate the effectiveness of OPAT. To the best of our knowledge, no such detailed analysis on OPAT has been undertaken before in Saudi Arabia. Nonetheless, outcomes are consistent with earlier research studies where overall effectiveness and cost benefits of the OPAT program have been reported and compared [15,16,17].

A home-based IV antibiotic treatment program is implemented in various countries [18,19]. However, very few non-private centers run such programs in Saudi Arabia and the Gulf region. Saudi health care system has witnessed a rise in demand for acute beds, while such a program is a convenient and safe alternative to managing acute beds [20]. The given therapy model offers a better practical option to patients. The literature also reflected that such programs have a high level of acceptance level in various countries, including the UK, Belgium, and other countries [21,22,23,24]. One study showed substantial differences in delivery route and model and infusion devices in different countries [24]. After reviewing two years of operation of a comprehensive OPAT program in the present study, our findings demonstrated that our OPAT program’s structure meets international guidelines [25]. The main advantage of this OPAT program is that it can be applied across a group of ages, including the elderly and children [22,23,26]. Moreover, this program helps to reduce the number of emergency department visits and incidents for many adverse events that arise due to conventional hospitalization [7]. This study highlighted that the OPAT program helped to save 1984 patient bed days during the study period. Thus, a comprehensive OPAT service would be helpful to shorten the length of hospital stay and increase the rate of discharge to home for patients with severe diseases [7,22,23]. Over the study period, the estimated medical cost reduction achieved with this program was 18 million SAR (approximately $5 million). This finding reflects that long-term administration of antimicrobial agents in the hospital is expensive and highlights the cost-effectiveness of the OPAT program [12,17,23,27]. The findings showed the potential of the OPAT program to render quality healthcare for the right patients. An estimated savings over 5 years from 57 OPAT services in the UK was found to be in the range of £60–77 million [22]. As oral therapies are the lowest cost treatment options, the cost can also be reduced if patients are switched from intravenous to oral therapies [28]. Nevertheless, comparing the results of these cost analyses of different countries is difficult due to the difference in OPAT models, healthcare systems, and cost analyses methods.

In our study, ertapenem was the most commonly prescribed antibiotic. However, high usage of these broad-spectrum antibiotics was due to the high prevalence of ESBL producing gram-negative organisms, as highlighted in previous studies [29,30]. Likewise, vancomycin was also commonly prescribed due to MRSA-based infections [31]. Urinary tract infections (UTI), osteomyelitis, and bacteremia were the most frequently treated infections in accordance with other OPAT programs [23,32]. The readmission rate in our study was 2.2% related to OPAT and 11.2% due to other than OPAT related issues. Other studies have described the rate of readmission between 3.6 and 27% [17,32,33,34,35].

Our study has some potential limitations. First, this was the retrospective observational study design. Secondly, the number of patients was relatively small, with the possibility of selection bias in testing the OPAT program’s effects on the Saudi Arabia health care system. The program was assessed in a single center, which limits its generalizability. However, this is the first study from Saudi Arabia which defines the efficacy and safety of the OPAT program. There was a lack of local OPAT guidelines and a national judicial framework. Presently, USA and UK already have national OPAT services guidelines, which ensure low risk and high-quality healthcare, but those guidelines cannot be generalized because of differences in the organization of health care in Saudi Arabia [11,25]. The development of national OPAT guidelines in Saudi Arabia can help to implement high-quality uniform OPAT care. Nevertheless, the program is considered successful in Saudi Arabia since the dropout rate and complication rate were very low. It offered a safe and comfortable alternative to managing patients in-house for similar cases. Other healthcare providers can run similar programs in Saudi Arabia. Hopefully, the positive results and findings from the program will encourage other healthcare service providers, clinicians, healthcare managers, and policymakers to create the necessary infrastructure to manage such programs across Saudi Arabia.

## 4. Methods

### 4.1. Study Design and Duration

We conducted a retrospective hospital-based observational study between November 2017 and March 2020.

### 4.2. Study Population

The participants were patients who presented at a single tertiary center in Saudi Arabia. The in-house OPAT team selected and evaluated all the patients who were discharged to their homes with parenteral antimicrobial treatment. An infectious disease specialist tested the patients’ clinical stability, which defined a systematic treatment plan, antibiotic and duration options. There was further monitoring of drugs, drug interactions, and dosing adjustment by the clinical pharmacist.

### 4.3. Inclusion Criterion

The other eligibility criteria included were: (1) being an adult (≥18 years old); (2) no history of any disorder related to acute psychiatric; (3) patients must be facilitated with home telephone and residential stay from the hospital; (4) patients must be stable enough not to require hospital admission other than for parenteral antimicrobials administration; (5) patients must be mentally stable with no history of drug addiction, and (6) no history of adverse reactions due to drugs for enrolled patients was monitored, including complications due to venous access and reactions due to allergies, nephrotoxicity, blockages, line fracture, slippage, and any other kind of discomfort or pain.

### 4.4. Data Collection

Demographic and clinical data were abstracted from the medical record, which included gender, age of the participant, site of infection, microorganisms that were isolated, diagnosis at a clinical level, antibiotic types, any past complications, duration, and readmission at a hospital for the treatment period, bed days saved, clinical outcomes, program outcomes, estimated cost reduction from our OPAT database and electronic medical records. The analysis report encompassed the sample results of collected microbiological data before any commencement of antibiotic regimens given to the patient during discharge. The evaluation of outcome was measured in terms of program safety, failure rate, and relapse rate. The primary outcome was the 30-days readmission rate of OPAT patients. Secondary outcomes were factors associated with OPAT failure. Readmission was defined as any unplanned hospital admission during the OPAT period within one month after OPAT completion. Patient bed days was calculated for all patients with or without OPAT.

### 4.5. Ethics Approval

Institutional Review Board approved study procedures and waived written informed consent.

### 4.6. Statistical Analysis

All analyses were performed with SPSS software. Only descriptive analysis was carried out for this observational study.

## Figures and Tables

**Figure 1 antibiotics-11-00441-f001:**
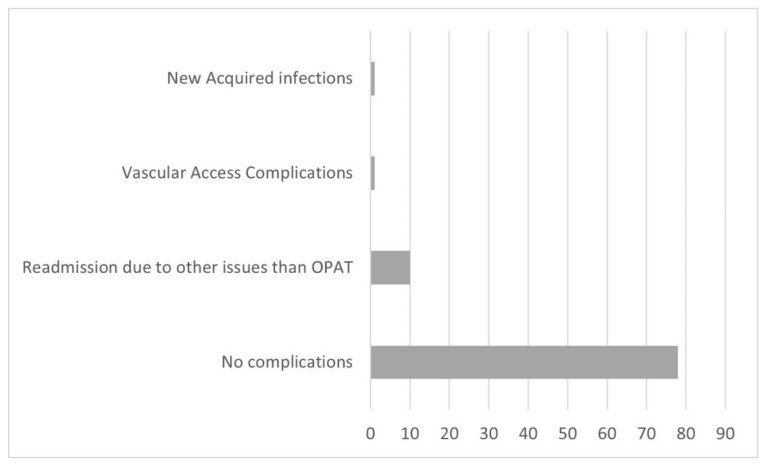
Distribution of patients according to treatment complications.

**Table 1 antibiotics-11-00441-t001:** Clinical diagnosis of patients enrolled in outpatient parenteral antimicrobial therapy.

Diagnosis	Number of Patients	Percentage (%)
Urinary tract infection	25	26.9
Osteomyelitis	16	17.2
Bacteremia (any source)	15	16.1
Endocarditis	6	6.5
Brucellosis	4	4.3
Tuberculosis	4	4.3
Hepatic microabscess	5	5.4
Diabetic foot	2	2.2
Pyomyositis	1	1.1
Respiratory infection	1	1.1
Septic arthritis	1	1.1
other	10	10.8

**Table 2 antibiotics-11-00441-t002:** Causative organisms in the patients treated with OPAT from November 2018 to March 2020.

Microorganisms	Number of Patients	Percentage (%)
ESBL *E. coli*	40	44.9
MRSA	15	16.9
MSSA	12	13.4
Brucella	4	4.5
*Mycobacterium tuberculosis*	4	4.5
Candida	3	3.4
ESBL *K. pneumoniae*	2	2.2
*Streptococcus mitis*	2	2.2
*Pseudomonas aeruginosa*	2	2.2
CRE *K. pneumoniae*	1	1.1
CRE *E. coli*	1	1.1
CoNS	1	1.1
*E. cloace*	1	1.1
*K. pneumoniae*	1	1.1
GNB	1	1.1

**Table 3 antibiotics-11-00441-t003:** Antibiotics selection preference among physicians.

Antibiotics	Number of Patients	Percentage (%)
Ertapenem	40	43.0
Vancomycin	11	11.2
Cloxacillin/Flucloxacillin	9	9.7
Gentamycin	4	4.3
Ceftriaxone	4	4.4
Anidulafungin	4	4.4
Cefepime	3	3.2
Meropenem	3	3.2
Amikacin	2	2.2
Cefazolin	2	2.2
Teicoplanin	2	2.2
Ampicillin/sulbactam	1	1.1
Ceftazidime/Avibactam	1	1.1
Flucloxacillin/Rifampin	1	1.1
Piperacillin/tazobactam	1	1.1
Rifampicin/Levofloxacin/Amikacin	1	1.1
Tigecycline	1	1.1

## Data Availability

The data presented in this study are available on request from the first/corresponding author.

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
