# Peer review of "Evaluate the Effectiveness of Outpatient Parenteral Antimicrobial Therapy (OPAT) Program in Saudi Arabia: A Retrospective Study"

_antibiotics, 2022, doi:10.3390/antibiotics11040441_

Round 1
Reviewer 1 Report
Overall the study design appears well done for retrospective design
Author Response
OPAT COMMENTS
Reviewer 1 Overall the study design appears well done for retrospective design
Author response: Thank you very much for your kind words – very much appreciated.
Reviewer 2 Report
Overall a fairly well-written and easy to read manuscript, but authors should consider the points below.
I have recommended "major revision" fundamentally because I do not understand how the authors derived their numbers (details below) and I feel they should be more explicit in their calculations. Thus, please consider major revisions as a safeguard at this point.
- Lines 27 and 28: how can the mean average of subject age and treatment duration have such huge errors? How was standard deviation calculated? Same (obviously) applies later in results (lines 88, 92-94). In my opinion it is not sufficient to indicate that "All analyses were performed with SPSS software (line 200).
- Line 124: "Delivery, and show substantial differences." Obviously an incomplete sentence. Please correct.
- Line 132: "...the OPAT program helped to save 1984 patient bed days during the study period" How was this calculation performed? It is not clear and no details are given.
- Line 140: "5757 OPAT services in the UK". The number is not correct. The reference indicates 57 OPATs. Obviously an honest/repetition mistake. Please correct.
- Section 4. "Methods" needs to be more detailed according to comments above.
- Finally, references are not as of journal requirement/format.
Author Response
Thank you so much for your review
OPAT COMMENTS
Reviewer 2 Overall a fairly well-written and easy to read manuscript, but authors should consider the points below.
Response: Thank you very much for your valuable comments.
I have recommended "major revision" fundamentally because I do not understand how the authors derived their numbers (details below) and I feel they should be more explicit in their calculations. Thus, please consider major revisions as a safeguard at this point.
- Lines 27 and 28: how can the mean average of subject age and treatment duration have such huge errors? How was standard deviation calculated? Same (obviously) applies later in results (lines 88, 92-94). In my opinion it is not sufficient to indicate that "All analyses were performed with SPSS software (line 200).
Response: Thank you very much for your valuable comments. We agree that this is a big variation. The difference is due to the inclusion of all ages in the hospital. The sample age is between 15 years old to 93 years old. The same goes for antibiotics treatment duration. We used SPSS for doing descriptive statistics only. We agree with the reviewer that these comments can be done using any other software like Excel.
- Line 124: "Delivery and show substantial differences." Obviously, an incomplete sentence. Please correct.
Response: Thank you very much for your valuable comments. We have removed this incomplete sentence.
- Line 132: "...the OPAT program helped to save 1984 patient bed days during the study period" How was this calculation performed? It is not clear, and no details are given.
Response: The calculation was performed by subtracting days of treatment on OPAT from total days of treatment. It is now updated in the method section
- Line 140: "5757 OPAT services in the UK". The number is not correct. The reference indicates 57 OPATs. Obviously an honest/repetition mistake. Please correct.
Response: Thank you very much for your valuable comments. We have made changes.
- Section 4. "Methods" needs to be more detailed according to comments above.
Response: Thank you very much for your valuable comments. We have made changes.
- Finally, references are not as of journal requirement/format.
Response: Thank you very much for your valuable comments. We have made changes.
Reviewer 3 Report
I believe this is an interesting study, but manuscript should be improved prior to publication. Besides English language improvement, my suggestions are below:
1) line 18 - the objective was, please write in past tense
2) shorten the abstract
3) line 57 - space is missing
4) line 63 - which center
5) line 66 - objectives were: to provide, to ensure etc
6) line 90 - no need for capital letters (diabetic, hepatic etc)
7) latin names in italic
8) line 101 - no need for TB abbreviation
9) line 105 - all numbers with 1 decimal place
10) table 4 could be presented as a figure. Moreover, it is unclear what was the intention of this table, describe further in results section
11) start discussion with main findings
12) line 132 - which study are you referring to? show results on bed days in the results section and compare with other studies in the discussion
Author Response
Thank you so much for your review
OPAT COMMENTS
Reviewer 3 I believe this is an interesting study, but manuscript should be improved prior to publication. Besides English language improvement, my suggestions are below:
Author response: Thank you very much for your kind words – very much appreciated.
1) line 18 - the objective was, please write in past tense
Response: Thank you very much for your valuable comments. We have made changes.
2) shorten the abstract
Response: Thank you very much for your valuable comments. We have made changes.
3) line 57 - space is missing
Response: Thank you very much for your valuable comments. We have made changes.
4) line 63 - which center
Response: Thank you very much for your valuable comments. We have made changes.
5) line 66 - objectives were: to provide, to ensure etc
Response: Thank you very much for your valuable comments. We have made changes.
6) line 90 - no need for capital letters (diabetic, hepatic etc)
Response: Thank you very much for your valuable comments. We have made changes.
7) latin names in italic
Response: Thank you very much for your valuable comments. We have made changes.
8) line 101 - no need for TB abbreviation
Response: Thank you very much for your valuable comments. We have made changes.
9) line 105 - all numbers with 1 decimal place
Response: Thank you very much for your valuable comments. We have made changes.
10) table 4 could be presented as a figure. Moreover, it is unclear what was the intention of this table, describe further in results section
Response: Thank you very much for your valuable comments. This table is about complications. We have made changes.
11) start discussion with main findings
Response: Thank you very much for your valuable comments. We have made changes.
12) line 132 - which study are you referring to? show results on bed days in the results section and compare with other studies in the discussion
Response: Thank you very much for your valuable comments. We have made changes.
Round 2
Reviewer 2 Report
Overall it appears to me that the authors have corrected what indicated by the reviewers and have replied adequately to the points mentioned by the same. I do not object to the publication of the article.
I just advise a careful final review by the editorial team in terms of text editing/minor corrections.